# Are Household Expenditures on Food Groups Associated with Children’s Future Heights in Ethiopia, India, Peru, and Vietnam?

**DOI:** 10.3390/ijerph17134739

**Published:** 2020-07-01

**Authors:** Sarah E. Weingarten, Kirk A. Dearden, Benjamin T. Crookston, Mary E. Penny, Jere R. Behrman, Debbie L. Humphries

**Affiliations:** 1Department of Epidemiology of Microbial Diseases, Yale School of Public Health, New Haven, CT 06510, USA; sweingarten11@gmail.com; 2IMA World Health, Washington, DC 20036, USA; kdearden@imaworldhealth.org; 3Department of Public Health, Brigham Young University, Provo, UT 84602, USA; benjamin_crookston@byu.edu; 4Instituto de Investigación Nutricional, La Molina 15024, Peru; mpenny@iin.sld.pe; 5Departments of Economics and Sociology, University of Pennsylvania, Philadelphia, PA 19104, USA; jbehrman@econ.upenn.edu

**Keywords:** household food group expenditures, height for age z-score (HAZ), longitudinal cohort study, food groups, expenditures on fats, Young Lives data

## Abstract

Household expenditure surveys, routinely conducted in low—and middle-income countries (LMICs), usually include questions pertaining to recent household expenditures on key food groups. When child anthropometrics are also available, such expenditure data can provide insights into household food purchasing patterns that are associated with subsequent child growth measures. We used data from 6993 children, born around 2001, from Ethiopia, India, Peru, and Vietnam, from the Young Lives younger cohort. We compared associations between two weeks of household food expenditures (in PPP—Purchasing Power Parity adjusted dollars) on food groups and child height-for-age-Z score (HAZ) at subsequent time points to assess longitudinal associations. Total food expenditures, rural/urban residence, maternal and paternal schooling, and child sex were included in our adjusted models because they may affect the relations between household food group expenditures and future child HAZ. In Ethiopia, India, and Peru every extra PPP$ spent on fats was associated with 0.02–0.07 higher future HAZ. In Vietnam every extra PPP$ spent on starches, was significantly associated with a 0.01 lower future HAZ. Across countries, different patterns of food expenditure and procurement may be differentially critical for predicting child HAZ. Our results demonstrate how expenditures on specific food groups can be associated with children’s linear growth. This study provides additional evidence of the utility of longitudinal household food expenditure data in understanding child nutritional status.

## 1. Introduction

According to UNICEF estimates from 2019, approximately 144 million children under the age of 5 years were stunted [1]. Poor childhood nutritional status has been associated with cognitive deficits, impaired growth, lower lifetime educational achievements and earnings, and lower birth weights in the next generation [2]. While child undernutrition is a complex and multifactorial problem, inadequate food intake is considered one of two immediate major causes of child undernutrition [3,4]. Unfortunately, information on actual food consumption is often lacking, making it difficult to identify which aspects of food intake are most closely associated with better future child nutritional status. However, household choices about allocation of the household budget on food in general, and particular food groups, may be an additional source of information about child food intake [5].

Current analytic methods used to examine the effects of diet on child nutritional status at a population-level face limitations in dietary intake data [6,7]. The “gold standard” for measuring current food consumption are observed-weighed food records (OWFR) and 24-hour recall (24 h) surveys [6]. These methods require careful training of interviewers, rigorous technical and managerial support, and may be unrealistic at the population-level in low- and middle-income countries (LMICs) [6]. Historically, the principle data source for assessing population-level food consumption and undernutrition has been the Food and Agriculture Organization’s (FAO) Food Balance Sheets (FBS). However, these data are limited to average individual consumption at the national level and provide no information on how food is distributed regionally, locally or within households [8]. Nor do they provide longitudinal data on child nutritional status.

Another data source addressing food is available to researchers, as governments in over 125 LMICs conduct regular (generally every 3–5 years) household consumption and expenditure surveys (HCES). These surveys include information on household expenditures on key food groups [7]. HCES have been validated against traditional gold-standard nutrition methods and were found to provide useful information on patterns of food access [9,10,11]. In addition, researchers note these surveys may be a good proxy for 24 h dietary measures in LMICs [10,12]. A subset of these data sets includes longitudinal information on child anthropometry. This subset may help identify patterns of food expenditure that are associated with children’s linear growth [5].

This study fills a gap in the existing literature about how different food group expenditures relate to children’s subsequent nutritional status. We hypothesize that household food expenditures on the seven key dietary diversity food groups—starches, fruits and vegetables, legumes, eggs, dairy, meat, and fat—will have differential associations with future child HAZ, after adjusting for child, parental, household, and community characteristics that we hypothesize may modify the relations between food group expenditures and child HAZ. Specifically, we investigate the associations of household food group expenditures at 5 and 8 years with child HAZ at 8 and 12 years, respectively.

## 2. Materials and Methods

Figure 1 represents the conceptual framework guiding this research. We posit that household food expenditures across food groups predict subsequent child height-for-age Z-scores (HAZ), low values of which (<−2 relative to international medians for well-nourished populations) are a measure of stunting (chronic undernutrition). We capitalize on three survey rounds of Young Lives’ younger-cohort longitudinal data to understand how household food group expenditures at earlier ages predict children’s HAZ subsequently, adjusting for other factors such as child sex, and household, parental, and community characteristics that might influence the relations between household food group expenditures and subsequent child HAZ.

### 2.1. Study Design and Participants

This study used data from the Young Lives (YL) younger cohort, a study of approximately 8000 children in Ethiopia, India, Peru, and Vietnam. In 2002, 2000 children were enrolled from each of the four countries and followed up in three more rounds when children were 5 years, 8 years and 12 years. The YL team used multistage sampling designs with the first stage consisting of a selection of 20 sentinel sites. Sampling was pro-poor. For example, in Ethiopia, the most food-insecure districts were oversampled as part of the study design. In Peru, the richest 5% of districts were excluded from the sample. While poor clusters were moderately oversampled, the final samples were diverse samples of social, geographic, and demographic groups. The sample in India consisted only of households from Andhra Pradesh (since split into Andhra Pradesh and Telangana), whereas the three other countries used nationwide samples. Sampling methods have been previously reported [2,5]. Additional study details are described elsewhere [13,14]. From age 1 year to age 12 years, the YL country cohorts lost between 1.5% and 5.7% of participants to attrition (Ethiopia 114/1999; India 81/2011; Peru 106/2052; Vietnam 36/2000). Household food expenditure data were collected starting with round 2. Consequently, this study included data from rounds 2–4, which were the only rounds available at the time this analysis was undertaken.

### 2.2. Study Indicators

#### 2.2.1. Household Food Expenditures

The respondent (usually the mother) was asked to report on consumption of food categories in the previous two weeks, ranging from 20 in Ethiopia, India and Vietnam, to 35 in Peru. For each food group, the respondent was asked to estimate the total monetary value of all foods in each of the food groups. To do this they were asked details of (a) expenditure on all individual foods in that food group consumed in the previous two weeks; (b) value of gifts received or food paid in lieu of wages in that food group; and (c) value of own stores used in that food group (which included foods from the household’s own production, shop or stocks). As previously reported [5], these food group expenditures were then aggregated to align with the seven food groups ((1) starches—bread, cereal, rice, grains and starchy vegetables; (2) fruits and vegetables; (3) legumes; (4) eggs; (5) dairy; (6) meat and fats; and (7) oils and oil seeds) in the WHO child dietary diversity measure [15]. At age 8 years, vitamin A-rich fruits and vegetables were recorded as separate categories as in the adult version of dietary diversity instrument [16,17]. Expenditures were adjusted to 2006 local currency and adjusted again to facilitate international comparability using purchasing power parity conversions (PPP) [5]. We summed total food expenditures across food groups and divided by the number of adult equivalents in the household to give total food expenditure per adult equivalent [18]. We winsorized expenditures in each food group to address outliers [19] for each of the four countries, replacing values below the 1st percentile with the 1st percentile, and replacing values above the 99th percentile with the 99th percentile. Expenditures for proteins, micronutrients, and animal source foods (ASF) were created by summing the appropriate categories to convey total expenditure on foods that contribute to those three food groupings. To create the protein expenditure variable, expenditures on legumes, meat, eggs, and dairy were summed. To create the micronutrient expenditure variable, expenditures on fruits and vegetables, legumes, meat and fish, eggs, and dairy were summed. To create the fat expenditure variable, expenditures on fats were summed. Finally, to create the ASF expenditure variable, expenditures on meat, eggs, and dairy were summed.

#### 2.2.2. Child Anthropometry

Field workers measured height using locally-made stadiometers with standing plates and moveable head boards accurate to 1 mm. Height-for-age Z-score (HAZ) was calculated using WHO 2006 standards for children 0–59 months [20] and WHO 2007 standards for older children [21]. Birth dates were recorded from children’s health cards when available, and from mothers’ reports otherwise. Our analyses focused on HAZ as the main indicator of chronic undernutrition in children. Children are classified as stunted if they have HAZ that are less than −2.0 [20,22].

#### 2.2.3. Control Variables

Other measures used to examine how certain characteristics might affect the relations between household food group expenditures and future child HAZ included children’s sex, rural/urban residence, total household food expenditures, and maternal and paternal schooling. Maternal and paternal schooling were coded as categorical variables for no schooling, 1–6 grades, 7–11 grades, and 12 or more grades of schooling.

#### 2.2.4. Country-Level Per-Capita Available Calories Data

These data were downloaded from Food Balance Sheets available on the Food and Agriculture Organization Corporate Statistical Database website [23]. Food supply (in kcal/capita/day) as well as protein and fat supply (in g/capita/day) were retrieved for each of the four study countries for each of the three study years. Food, protein, and fat supplies were also broken out by source: either animal or vegetable products. These data were used to assess the heterogeneity in country-level food availability.

### 2.3. Statistical Methods

We used SAS software (Version 9.4. Copyright 2012, SAS Institute Inc., Cary, NC, USA) for all analyses. Results were considered statistically significant for *p*-values < 0.05. Statistical differences in food group expenditures by country, round, and rural/urban residence were assessed using the Wilcoxon Rank Sum test, and medians and interquartile ranges were reported to highlight the differences in expenditure patterns. Means and 95% confidence intervals (for continuous variables) and frequencies and percentages (for categorical variables) were reported by country and round for all other key study indicators. We present associations between household food group expenditures and child HAZ using single and multivariable ordinary least squares regressions, controlling for total food expenditures, rural/urban residence, maternal and paternal schooling, and sex of child. Our models assessed how expenditures at 5 years predict HAZ 8 years, and also how expenditures at 8 years predict HAZ at 12 years.

### 2.4. Ethical Review

The secondary analysis of de-identified public use data in this study was identified as exempt by the University of Pennsylvania Institutional Review Board under protocol number #814690.

## 3. Results

### 3.1. Country-Level Food Energy and Macronutrient Availability

Heterogeneity in the country-level food energy and macronutrient availability context is important to take into consideration as a backdrop to the additional country-level information and comparisons provided in this study. Based on FAOSTAT country-level food balance sheets, caloric availability varies across the four countries and rounds, with Ethiopia consistently showing the lowest overall energy intake and Vietnam consistently showing the highest (Figure 2). In addition, Vietnam has the highest consumption of animal products, and Ethiopia has the lowest.

### 3.2. Descriptive Characteristics

The child growth characteristics of these populations (Table 1) have been characterized extensively in previous publications and our analyses are consistent with previous reports [2,5,24]. Mean HAZ was negative in all countries in each round, and mean HAZ was less negative at 12 years than at 5 years in India (−1.65 to −1.44), Peru (−1.53 to −1.02) and Vietnam (−1.35 to −1.06), although it remained constant for Ethiopia (−1.48 to −1.47) (Table 1). The percentages of individuals who lived in rural areas in each country remained relatively constant in Ethiopia (59%, 60%), India (72–74%) and Vietnam (80%, 81%) across the three time periods included in the study. In contrast, Peru experienced greater rural to urban migration as demonstrated by a decrease from 44% rural residence in Round 2 to 27% in Round 4. The majority of mothers in Ethiopia (51%) and India (51%) reported no schooling. The proportion of mothers reporting completing 7–12 years of schooling was highest in Peru (42%) and Vietnam (47%). In Ethiopia, fathers were more likely to report schooling levels of 1–6 grades (46%), while in India (36%), Peru (49%), and Vietnam (53%), fathers were more likely to report schooling levels in the range of 7–12 grades (Table 1).

### 3.3. Differences by Rural/Urban Residence

#### 3.3.1. Ethiopia

When assessing the differences in 15-day household expenditure on key food groups by rural and urban residence (Table 2), urban households had significantly higher expenditures on starches, fruits and vegetables, meat, legumes, eggs, and fats compared to rural household across all rounds, with the exception of expenditures on eggs at 8 years. Dairy expenditures across all rounds did not differ by rural/urban residence with mean expenditures of less than 0.005 PPP$ for both groups. In both urban and rural settings, households spent the greatest amount on starches ($7–10/15 days), with the median expenditure for every other food group except urban legumes less than $1.0/15 days.

#### 3.3.2. India

Urban households generally had significantly higher expenditures on fruits and vegetables, dairy, and fats per adult equivalent compared to rural households, with the only exception being expenditures on starches at 5 years, for which rural households spent a significantly greater amount (Table 2). Expenditures on meat were not statistically different between rural and urban households at age 5 years and 12 years, but data at 8 years showed significantly larger expenditures on meat by rural households. With regard to eggs, a significant difference in expenditure by rural/urban residence was detected at 5 years, with urban households having larger expenditures on eggs. Legumes were the only food group significantly associated with greater expenditure by rural households at each age. Starches had the greatest absolute expenditure in both urban and rural households ($5–7/15 days).

#### 3.3.3. Peru

Relative to rural households, urban households tended to expend significantly more resources across all rounds on fruits and vegetables, meats, and dairy. (Table 2). Rural households had greater expenditures on starches, legumes, and fats compared to urban households, with the only exceptions being expenditures on starches and legumes at 5 years, which were not significantly different between rural and urban households. Differences in expenditures on eggs varied across all rounds. The greatest absolute expenditures in rural and urban households in Peru were on starches ($13–18/15 days), with meats having the second greatest expenditures ($5–11/15 days).

#### 3.3.4. Vietnam

Urban residents had significantly larger expenditures than rural residents on fruits and vegetables, meat, legumes, eggs, dairy, and fats across all three rounds (with the exception of expenditures on legumes at 8 years, which were not significantly different by rural/urban status) (Table 2). Expenditures on starches at 5 years and 8 years were significantly greater for rural households, while at 12 years, rural households had significantly smaller expenditures. The greatest absolute expenditures by rural and urban households were on meat ($9–21/15 days), with starch expenditures having the second largest absolute expenditures ($8–12/15 days).

### 3.4. Unadjusted and Food Group-Only Models

#### 3.4.1. Ethiopia

Each of the 10 food group expenditures (fruits and vegetables, legumes, meat, eggs, dairy, starches, fats, proteins, micronutrients, and ASF) at 5 years had a significant unadjusted association with HAZ at 8 years (Table 3a). The unadjusted fats model had the highest adjusted R^2^ value, and the unadjusted coefficient suggested every $1 increase expenditure on fats at 5 years was associated with a 0.15 increase in HAZ at 8 years. The models for protein, micronutrient, and ASF expenditures (models included expenditures on all food groups encompassed by the aggregate group), showed that after including other food groups that provide similar nutritional components, only expenditures on legumes and fruits and vegetables at 5 years were significantly positively associated with HAZ at 8 years. Of those with significant univariate associations, fats had the highest R^2^ value (0.0183), and the largest coefficient (0.16). The models for proteins, micronutrients, and ASF expenditures suggest that meats, rather than legumes were the primary driver of the associations between protein and HAZ at 12 years. Before and after adjusting for other food groups, every additional $1 spent on meat at 8 years was associated with a 0.04 increase in HAZ at 12 years. Fruits and vegetables continued to have a significant association with future HAZ after including other micronutrient-rich food groups, suggesting that every extra $1 spent on fruits and vegetables at 8 years was associated with a 0.12 increase in future HAZ.

#### 3.4.2. India

Meat, dairy, fats, proteins, micronutrients, and ASF expenditures at 5 years all had significant associations with HAZ at 8 years (Table 3b). Similar to Ethiopia, expenditures on fats had the largest association with future HAZ for every extra $1 spent (β: 0.11), but expenditures on dairy at 5 years were associated with the greatest variation in HAZ at 8 years (adj R^2^: 0.0405). After adjusting for protein, micronutrient, and ASF food group expenditures in three separate models, dairy and meat expenditure continued to be significantly associated with HAZ at 8 years. Every additional $1 spent on dairy was associated with a 0.10 increase in future HAZ, while every additional $1 spent on meat was associated with approximately a 0.02 increase. When looking at food group expenditures at 8 years, dairy and fats continued to be significantly associated with future HAZ and with relatively high coefficients (dairy: 0.10; fats: 0.14) and R^2^ values (dairy: 0.0473; fats: 0.0212). Even after including other food group expenditures in the proteins, micronutrient, and ASF models, dairy expenditures were still significantly associated with HAZ at 12 years.

#### 3.4.3. Peru

Expenditures on all food groups at 5 years except for legumes, starches, and fats had significant associations with HAZ at 8 years (Table 3c). Of all the associations, expenditure on eggs had the largest coefficient (0.09), while total expenditures on dairy had the highest adjusted R^2^ value (0.0718). After including the other food groups in the three composite models, the association between egg expenditures at 5 years and future HAZ was not significant. Interestingly, in both the protein and micronutrient analyses, legume expenditure was associated with a 0.04 decrease in HAZ at 8 years for every extra $1 spent. Fruits and vegetables, meat, and dairy expenditures were still significantly associated with HAZ at 8 years (with the exception of meat expenditures in the micronutrient model, which was only marginally significant). Similar patterns were noted for the analyses assessing relationship of expenditures at 8 years to HAZ at 12 years.

#### 3.4.4. Vietnam

All food group expenditures at both 5 years and 8 years were significantly associated with future HAZ except for expenditures on starches (Table 3d). When investigating individual food groups rather than the composite protein, micronutrient and ASF expenditures, expenditures on meat, dairy, and fruits and vegetables at 5 years remained significantly associated with HAZ, while at 8 years expenditures on meat, eggs, fruits and vegetables remained significantly associated with future HAZ. For expenditures at 5 years, we found that fats had the largest association with higher HAZ for every dollar spent (β = 0.19), but micronutrient expenditures were associated with the most variation in HAZ at 8 years (R^2^ = 0.097). This micronutrient pattern was also observed with expenditures at 8 years, but instead of fats, the food group with the greatest association with future HAZ per dollar spent was eggs.

### 3.5. Adjusted Models Investigating Association of Key Food Expenditures and Future HAZ

Following preliminary analyses reported in Table 2 and Table 3 individual food groups were chosen for detailed review due to the relatively large accompanying coefficients and/or significance of these models compared to other variables (see Figure 3a–d; models for other food groups can be found in Appendix A). Expenditures on fats at 5 years and 8 years, and how they were associated with future HAZ after adjusting for other key covariates are discussed for Ethiopia (Table 4a), India (Table 4b), and Peru (Table 4c). In Vietnam (Table 4d), expenditures on starches at 5 years and 8 years are highlighted due to the relative significance of the final, adjusted association compared to other food groups (models for other food group expenditures in Vietnam can be found in Appendix A.

#### 3.5.1. Ethiopia

Fat expenditures had the highest coefficient in univariate models at both 5 years and 8 years (Table 4a, Models 1a and 1b). However, in models that included individual, household and community covariates fat expenditures were no longer significantly associated with future HAZ. Additionally, none of the models for individual food groups that adjusted for covariates had a significant association between the analyzed food expenditure group and child HAZ score at either round (See Appendix A). When Total Food Expenditures were included, the individual food group expenditures were no longer significantly associated with future child HAZ. Fat expenditures, while not significant after adjusting for sex, rural/urban residence, total food expenditures, and maternal and paternal schooling, retained the highest coefficients compared to other food groups (Appendix A).

After adjusting for the five aforementioned covariates, every additional $1 spent on fats at 5 years was associated with a 0.03 increase in HAZ at 8 years and each additional $1 spent on fats at 8 years was associated with a 0.02 increase in HAZ at 12 years. Child sex, rural/urban residence, total food expenditure, maternal schooling (except for mothers with 12 or more grades of schooling), and paternal schooling at 5 years were significantly associated with HAZ at 8 years, and individual food group expenditures were not. Children living in rural areas at 5 years had HAZ at 8 years that were, on average, lower by 0.19 compared to children living in urban areas. Additionally, parental schooling (1–12 grades for mothers and 1–12 or more grades for fathers) was associated with a 0.14–0.26 higher child HAZ at 8 years compared to those whose parents had no schooling. At 8 years, the only covariates that were significantly associated with HAZ at 12 years were rural/urban residence, and maternal schooling (except for mothers with 12 or more grades of schooling).

#### 3.5.2. India

Adjusted expenditures on fats at 5 years and 8 years were significantly associated with future HAZ (Table 4b). After adjusting for sex, rural/urban residence, total food expenditures, and maternal and paternal schooling, every additional $1 spent on fats at 5 years was associated with a 0.07 increase in HAZ at 8 years. Of the variables included in the adjusted model at 5 years (Table 4b, Model 2a), all were significantly associated with HAZ at 8 years except for total food expenditures and paternal schooling of 1–6 grades. This model had a relatively high adjusted R^2^, with 12.2% of variation in HAZ at 8 years predicted. Expenditures on fats at 8 years showed a similar relationship with HAZ at 12 years (Table 4b, Model 2b), with every additional $1 spent on fats at 8 years associated with a 0.06 increase in 12 years HAZ. In the adjusted model at 12 years, however, compared to the adjusted model at 8 years (Table 4b, Model 2a), sex and several levels of maternal schooling were no longer significantly associated with future HAZ.

#### 3.5.3. Peru

Expenditures on fats were significantly associated with HAZ at 8 years (Table 4c). After adjusting for all control variables, every extra $1 spent on fats at age 5 years was associated with a 0.05 increase in HAZ at 8 years (Table 4c, Model 2a), while every extra $1 spent on fats at 8 years was associated with a 0.04 increase in child HAZ at 12 years (Table 4c, Model 2b). Paternal schooling was not significantly associated with future child HAZ after adjusting for all other variables in the model.

#### 3.5.4. Vietnam

Starches were the only significant food group expenditure associated with future HAZ at both 5 years and 8 years (Table 4d). Greater expenditure on starches was negatively associated with future HAZ. At age 5 years and 8 years, every extra $1 spent on starches, was significantly associated with a 0.01 decrease in future HAZ after adjusting for covariates (Table 4d, Models 2a and 2b). All covariates in both of these adjusted models, except for sex, were significantly associated with future HAZ. Maternal schooling had the largest estimated coefficients. Most notably, Vietnamese children with mothers who had 12 or more years of schooling, on average, had future HAZ scores that were 0.92–0.96 higher compared to children with mothers who had no schooling. These models had some of the highest R^2^ values reported for any of the food group models across the four countries. The adjusted model of starch expenditures at 5 years was associated with approximately 20% of the variation in future HAZ and the adjusted model of starch expenditures at 8 years was associated with approximately 17.5% of the variation in future HAZ.

## 4. Discussion

We used data on 6.993 children born around 2001 from Ethiopia, India, Peru, and Vietnam collected at ages 5, 8, and 12 years from the Young Lives younger cohort to assess the longitudinal associations between expenditures on key food groups at 5 years and 8 years (including starches, fats, legumes, meat, fruits and vegetables, eggs, dairy, animal source food, proteins, and micronutrients) and child HAZ at ages 8 years and 12 years, respectively. While there was much heterogeneity among the countries in terms of food energy and macronutrient availability, several cross-country similarities emerged from our analysis. Across three of the four countries, household expenditures on fats were associated with the largest changes in future child HAZ before and after adjusting for children’s sex, rural/urban residence, total food expenditures, and parental schooling.

Expenditures on animal source foods, proteins, and micronutrient rich foods had very small associations with future HAZ. Increased expenditures on starches were associated with lower HAZ in both Vietnam and Peru in unadjusted and adjusted models. Of all the adjusted Vietnam models, the associations of HAZ with starches at 5 years and 8 years were the only ones that reached significance. The addition of total food expenditures to models with expenditures on specific food groups dramatically altered patterns of significance, particularly in Ethiopia. Parental schooling of 7–12 grades or more was an important predictor of higher HAZ across all of the countries, with the exception of paternal schooling in Ethiopia at 8 years and Peru at both 5 years and 8 years. Finally, urban residence was associated with higher HAZ in every country.

India showed variations in significance of unadjusted associations between food groups and future HAZ across study ages. After adjusting for children’s sex, rural/urban residence, total food expenditure, maternal schooling, and paternal schooling, expenditure on fats was the only food group in India significant across both ages, and also had the greatest magnitudes of associations compared to the other food groups. While a few studies have linked fat expenditures in India to higher stature, there are few studies demonstrating a link between fat consumption and child HAZ [24]. One study identified a co-movement of declining calorie, protein, and fat intake for both adults and children and a corresponding lack of progress in improving nutritional outcomes for children [25]. Another study, conducted in South Africa, found that intake of fat, along with calcium, phosphorous, vitamin D, riboflavin, and vitamin B12 (nutrients that typically occur in milk), was significantly lower in stunted children compared to non-stunted children [26]. A third study found that diets that provide < 22% of energy from fat and that are low in animal fats may restrict growth [27].

In Vietnam, after adjusting for sex, rural/urban residence, total food expenditure, maternal schooling, and paternal schooling, starches were the only food group expenditure category that was significantly associated with future HAZ at both age 5 years and 8 years. The findings regarding starch expenditures in Vietnam were consistent with several other studies that concluded that greater expenditures on rice and grains increased the odds and prevalence, respectively, of child stunting [28,29]. Consumption of starches has been associated with lower dietary diversity, particularly in poor populations in low-income countries, where diets are often unbalanced and composed primarily of starchy staples [30]. There is a large body of research from various countries linking dietary diversity to child growth [30,31,32,33,34].

In Ethiopia the expenditures on specific food groups were not significant when total food expenditure was included in the model, although this was not consistently the case in other countries. When both total food expenditure and expenditures on fat were included in the model in India and Peru (at age 5 years and 8 years), expenditures on fat remained significant. This suggests that total resources spent on food may be more important than expenditures on specific food groups in Ethiopia. Total food expenditures are considered a proxy for household income [32], so this pattern may point to household income and total food availability as key limiting factors for child HAZ in Ethiopia.

Higher parental schooling (both maternal and paternal) was generally associated with higher future HAZ in children, although there were several exceptions. In Ethiopia the highest level of maternal schooling (greater than 12 grades) was not significantly associated with future HAZ at either age and no levels of paternal schooling was associated with HAZ at 12 years. Peruvian households showed no significant association between paternal schooling and future HAZ across all food groups. The associations of mothers’ schooling with child health and nutrition has been well documented [4,35,36], as well as the significance of fathers’ schooling in predicting lower probabilities of stunting [37]. Mothers, though, are more likely than fathers to allocate family resources in ways that promote their children’s nutrition [38], which may help to explain the insignificant paternal schooling associations.

Children residing in urban areas had higher future HAZ compared to children living in rural areas in all four countries. In the final adjusted models for all food groups across all countries, urban residence was associated with a 0.19 to 0.47 increase in future HAZ. These magnitudes of association were second only to those for parental schooling and are consistent with the large body of literature that shows rural children have worse health outcomes compared to their urban-dwelling peers [4,39,40,41].

There are several cross-sectional studies that have demonstrated significant associations between household food expenditures and concurrent child nutritional status. A study conducted using nationally representative survey data from 2007 in Indonesia found that a higher mean proportion of household expenditure on soybeans was significantly associated with lower odds of being wasted (low weight-for-height) and underweight (low weight for age) [42]. Additionally, a lower mean proportion of household expenditure on sugar and cooking oil was significantly associated with lower odds of being wasted or underweight [42]. Another more recent study, also from Indonesia, examined associations between dietary diversity and stunting, but found an unadjusted association between food group expenditures (food groups not specified) and stunting [34]. This was the only significant independent variable not directly assessing food or nutrient consumption used in the study to demonstrate a significant association with child stunting [34]. Research conducted using nutritional surveillance data from 2000 to 2005 in Bangladesh found that for children aged 6–59 months, household expenditure on rice was significantly associated with increased odds of stunting, and expenditure on non-rice foods was significantly associated with decreased odds of stunting [28]. Thus, households in higher quintiles of expenditures on non-rice foods and lower quintiles of expenditures on rice had a lower prevalence of child stunting [28]. Finally, a study conducted using nutritional surveillance data from 1999–2003 in Indonesia found that rural households and urban households drawn from poor communities that spent a greater proportion of their resources on animal source and non-grain foods had a lower prevalence of child stunting [29].

Several previous supplementation trials demonstrated the impact of lipid-based nutrient supplementation in young children on future HAZ, which is likely due in large part to increased caloric intake [43,44,45]. Another possible pathway is that fats may be associated with children’s linear growth due to their importance in ensuring the bioavailability of vitamins such as vitamin A and vitamin D [46,47]. However, studies on dietary fat intake in school-age children are primarily focused on overweight and the reduction of fat intake [48,49]. Our analyses suggest that dietary fat intake may still be a limiting factor for linear growth in some populations.

### Limitations

There are several limitations to this study. First, the YL dataset does not provide information on how purchased foods are distributed within the household so that the actual quantity of food each child consumed is unknown. This study focused primarily on how food expenditures were associated with future child height, but there is an extensive literature highlighting the importance of non-food influences on child height [50,51,52,53,54,55]. Many of our models had relatively low R^2^ values, which speaks to the complexity of child growth and the multiplicity of influences on child HAZ over time. We were limited by the food categorizations used for collecting food expenditure data. Fat expenditures were limited to oil and oil seeds. Other food groups, including dairy (butter, cheese, whole milk products), animal products, and legumes, nuts and seeds (nuts) have additional fats. Thus, the analysis was not able to isolate fat expenditures completely. While we cannot attribute a direct effect of increased spending on oils to an increase in linear growth the results provoke exploration of possible pathways or associations that may explain this robust association. For instance, does increased spending on oil mean increased consumption of higher energy density foods? Further understanding of this association might help us understand more about how dietary patterns that go beyond food ingredients may influence linear growth. Additionally, our analyses using expenditures at any given age (5 years or 8 years) did not consider patterns of spending before or after that age. For this analysis we used expenditures at each age to assess associations with HAZ at the next observed age. This analysis focused on linear growth as the outcome. Future work should incorporate analysis of BMI as an important indicator of healthy growth, given the growing concerns about BMI and risk of obesity in children in countries such as Peru.

## 5. Conclusions

Our results suggest different facets of food expenditure and procurement may be important for future child HAZ in different countries. In Ethiopia, results pointed to total food expenditure being a critical and significant component associated with subsequent child heights, though expenditures on fats had larger associations with subsequent child heights than other food groups. In India and Peru, expenditure on one specific food group—fats—was significantly positively associated with future HAZ. Finally, in Vietnam, starches were significantly negatively associated with future HAZ, which may relate to the importance of dietary diversity for this country. Stunting and, in general, child undernutrition are complex and multifaceted problems that food expenditure and subsequent consumption can only partly explain. Our results are useful in that they pinpoint specific facets of the diet that may have had the greatest impact on child growth for each country. They also point to the importance of examining context, and caution about generalizing from one country’s situation to others. These results highlight the usefulness of longitudinal household food expenditure survey data that include measures of children’s anthropometrics in analyzing children’s nutrition at the country level. They further provide useful information for child food and nutrition policies in these four countries.

## Figures and Tables

**Figure 1 ijerph-17-04739-f001:**
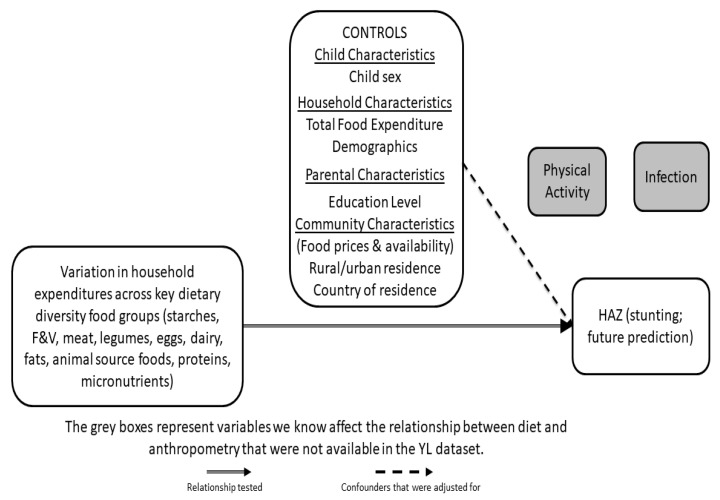
Conceptual Framework: This figure highlights the relationship between child height (height-for-age Z score (HAZ) far right) and previous household expenditures on specific dietary food groups (far left), while controlling for key child, household and community characteristics (upper box). YL = Young Lives.

**Figure 2 ijerph-17-04739-f002:**
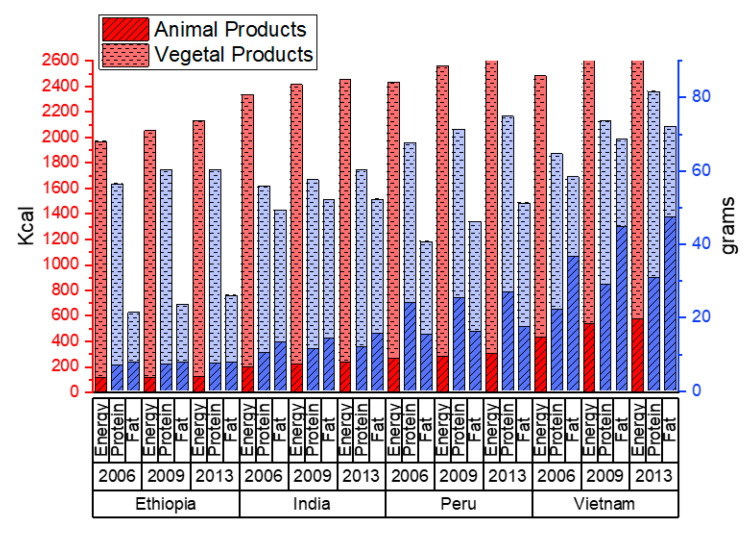
We present here the per capita availability of energy (kcal, left axis), protein and fat (grams, right axis) from animal and vegetable sources, in each of the four countries (Ethiopia, India, Peru, Vietnam) and each of the three years (2006, 2009, 2013) covered by this study. Per capita availability is taken from the food balance sheets of the Food and Agricultural Organization Statistical database [23].

**Figure 3 ijerph-17-04739-f003:**
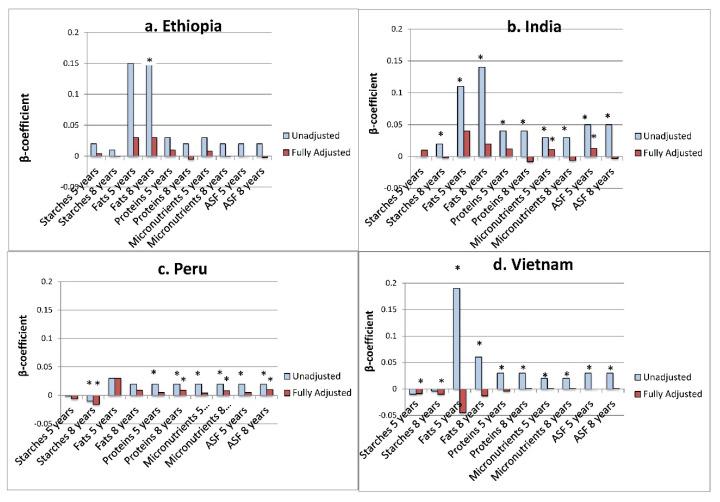
Unadjusted and adjusted coefficients for food groups in association with HAZ at next age of measurement for each country (**a**–**d**). We present here the unadjusted (blue) and adjusted (red) coefficients for each food group in bivariate and multivariate regression analyses with HAZ at the next age of measurement (i.e., 8 years HAZ and 5 years food group expenditures; 12 years HAZ and 8 years food group expenditures). * = *p* value < 0.05.

**Table 1 ijerph-17-04739-t001:** Descriptive Statistics of Key Study Variables by Country and Child Age at Time of Survey.

Ethiopia	5 Years (Round 2)	8 Years (Round 3)	12 Years (Round 4)
	*n* = 1744	*n* = 1742	*n* = 1733
HAZ, mean (95% CI)	−1.48 (−1.53, 1.43)	−1.22 (−1.27, −1.17)	−1.47 (−1.52, −1.42)
Female, *n* (%)	815 (46.7)	814 (46.7)	809 (46.7)
Rural Residence, *n* (%)	1047 (60.0)	1046 (60.1)	1028 (59.3)
Maternal Schooling, *n* (%)			
No schooling	887 (51.3)	886 (51.3)	884 (51.4)
1–6 years	521 (30.1)	520 (30.1)	520 (30.3)
7–12 years	290 (16.8)	290 (16.8)	284 (16.5)
12+ years	32 (1.9)	32 (1.9)	31 (1.8)
Paternal Schooling, *n* (%)			
No schooling	432 (25.8)	431 (25.8)	431 (25.9)
1–6 years	768 (45.9)	767 (45.9)	768 (46.2)
7–12 years	372 (22.3)	372 (22.3)	365 (21.9)
12+ years	100 (6.0)	100 (6.0)	100 (6.0)
**India**	**5 Years (Round 2)**	**8 Years (Round 3)**	**12 Years (Round 4)**
	***n* = 1804**	***n* = 1806**	***n* = 1801**
HAZ, mean (95% CI)	1.65 (−1.70, −1.61)	−1.46 (−1.50, -1.41)	−1.44 (−1.49, −1.39)
Female, N (%)	842 (46.7)	842 (46.6)	837 (46.5)
Rural Residence, N (%)	1337 (74.1)	1324 (73.3)	1299 (72.1)
Maternal Schooling, N (%)			
No schooling	914 (50.8)	916 (50.8)	913 (50.8)
1-6 years	363 (20.2)	363 (20.1)	363 (20.2)
7-12 years	475 (26.4)	475 (26.4)	473 (26.3)
12+ years	48 (2.7)	48 (2.7)	48 (2.7)
Paternal Schooling, N (%)			
No schooling	591 (32.8)	593 (32.9)	592 (32.9)
1-6 years	435 (24.2)	435 (24.1)	434 (24.1)
7-12 years	647 (35.9)	647 (35.9)	644 (35.8)
12+ years	128 (7.1)	128 (7.1)	128 (7.1)
**Peru**	**5 Years (Round 2)**	**8 Years (Round 3)**	**12 Years (Round 4)**
	***n*** **= 1795**	***n*** **= 1788**	***n*** **= 1775**
HAZ, mean (95% CI)	−1.53 (−1.58, −1.48)	−1.15 (−1.20, −1.10)	−1.02 (−1.07, −0.97)
Female, *n* (%)	896 (49.9)	896 (50.1)	883 (49.8)
Rural Residence, *n* (%)	795 (44.3)	498 (27.9)	471 (26.5)
Maternal Schooling, *n* (%)			
No schooling	149 (8.4)	149 (8.4)	145 (8.2)
1–6 years	640 (35.9)	637 (35.9)	634 (36.0)
7–12 years	751 (42.2)	748 (42.2)	744 (42.3)
12+ years	241 (13.5)	240 (13.5)	238 (13.5)
Paternal Schooling, *n* (%)			
No schooling	24 (1.4)	24 (1.4)	24 (1.4)
1–6 years	553 (31.8)	552 (31.9)	546 (31.7)
7–12 years	855 (49.2)	850 (49.1)	846 (49.2)
12+ years	307 (17.7)	306 (17.7)	305 (17.7)
**Vietnam**	**5 years (Round 2)**	**8 years (Round 3)**	**12 years (Round 4)**
	***n*** **= 1788**	***n*** **= 1754**	***n*** **= 1684**
HAZ, mean (95% CI)	−1.35 (−1.40, −1.30)	−1.11 (−1.16, −1.06)	−1.06 (−1.11, −1.00)
Female, n (%)	871 (48.7)	854 (48.7)	818 (48.6)
Rural Residence, n (%)	1435 (80.3)	1418 (80.8)	1369 (81.3)
Maternal Schooling, n (%)			
No schooling	181 (10.2)	176 (10.1)	177 (10.6)
1–6 years	637 (35.9)	627 (36.0)	611 (36.6)
7–12 years	832 (46.9)	816 (46.9)	771 (46.2)
12+ years	124 (7.0)	121 (7.0)	111 (6.7)
Paternal Schooling, n (%)			
No schooling	119 (6.8)	113 (6.6)	114 (6.9)
1–6 years	550 (31.6)	543 (31.7)	525 (31.9)
7–12 years	926 (53.1)	910 (53.2)	871 (53.0)
12+ years	148 (8.5)	145 (8.5)	135 (8.2)

**Table 2 ijerph-17-04739-t002:** Differences in 15-day household food expenditure (adjusted PPP) by age on key dietary diversity food groups by rural/urban status.

Country and Food Groups	5 Years-Rural	5 Years-Urban	8 Years-Rural	8 Years-Urban	12 Years-Rural	12 Years-Urban
Ethiopia	Median (IQR ^1^)	Median (IQR)	Median (IQR)	Median (IQR)	Median (IQR)	Median (IQR)
Starches	7.67 (7.49)	9.44 (7.88) ***	8.39 (8.11)	10.30 (7.95) ***	8.55 (7.93)	10.59 (6.08) ***
FV	0.36 (0.74)	0.53 (0.78) ***	0.29 (0.75)	0.51 (0.73) ***	0.41 (0.63)	0.87 (1.19) ***
Meat	0.00 (0.52)	0.00 (1.46) **	0.00 (0.00)	0.00 (1.43) ***	0.00 (0.38)	0.00 (2.77) ***
Legumes	0.80 (1.65)	1.16 (1.46) ***	0.86 (1.17)	0.92 (0.96) **	0.83 (1.11)	1.27 (1.38) ***
Eggs	0.00 (0.00)	0.00 (0.28) **	0.00 (0.22)	0.00 (0.33)	0.00 (0.23)	0.00 (0.56) ***
Dairy	0.00 (1.14)	0.00 (1.18)	0.00 (1.16)	0.00 (1.05)	0.00 (1.38)	0.00 (1.38)
Fats	0.73 (0.81)	1.38 (1.50) ***	0.57 (0.59)	1.10 (1.11) ***	0.57 (0.53)	1.30 (1.25) ***
**India**						
Starches	7.03 (4.64)	5.39 (4.24) ***	5.27 (5.26)	6.22 (5.24) **	5.17 (5.12)	6.04 (5.20) *
FV	2.52 (2.11)	2.79 (2.28) **	3.21 (2.58)	3.59 (2.66) ***	3.57 (2.42)	4.05 (2.58) ***
Meat	1.98 (2.97)	2.05 (2.77)	2.29 (2.70)	2.17 (2.82) **	2.19 (2.45)	2.15 (3.09)
Legumes	1.21 (1.00)	0.87 (0.71) ***	1.44 (1.22)	1.32 (1.07) **	1.02 (0.81)	0.95 (0.74) **
Eggs	0.31 (0.57)	0.37 (0.39) *	0.31 (0.52)	0.35 (0.54)	0.30 (0.52)	0.30 (0.50)
Dairy	0.91 (2.12)	2.32 (2.30) ***	0.89 (1.81)	2.96 (3.32) ***	1.36 (2.18)	2.90 (2.72) ***
Fats	1.66 (1.01)	1.73 (1.05)	1.50 (1.19)	1.83 (1.35) ***	1.56 (1.18)	1.82 (1.19) ***
**Peru**						
Starches	13.59 (8.94)	13.37 (7.33)	16.83 (9.65)	13.56 (7.36) ***	17.95 (11.98)	14.19 (8.45) ***
FV	1.80 (2.05)	2.96 (3.25) ***	2.96 (3.15)	3.36 (3.65) **	3.93 (3.87)	4.57 (4.86) ***
Meat	4.51 (7.66)	8.53 (8.20) ***	7.54 (9.60)	9.63 (9.08) ***	9.71 (11.14)	11.13 (9.75) ***
Legumes	1.25 (1.75)	1.13 (1.28)	1.78 (1.97)	1.24 (1.42) ***	2.06 (2.25)	1.36 (1.54) ***
Eggs	0.77 (1.04)	1.05 (1.12) ***	1.03 (1.21)	1.17 (1.14)	1.53 (1.64)	1.38 (1.29) **
Dairy	2.17 (3.82)	5.25 (6.50) ***	3.18 (4.54)	4.48 (5.26) ***	4.04 (4.48)	5.02 (5.68) ***
Fats	1.29 (1.16)	1.18 (0.91) ***	1.69 (1.33)	1.39 (1.08) ***	1.69 (1.73)	1.25 (1.34) ***
**Vietnam**						
Starches	8.90 (4.41)	8.11 (4.74) ***	9.97 (5.46)	8.78 (5.75) ***	10.82 (6.91)	11.53 (10.04) *
FV	3.12 (3.02)	4.89 (5.87) ***	3.61 (3.40)	6.69 (6.47) ***	4.05 (4.28)	7.60 (8.07) ***
Meat	8.90 (8.01)	12.30 (11.15) ***	10.19 (8.81)	15.97 (12.74) ***	13.68 (11.93)	21.07 (19.72) ***
Legumes	0.00 (0.30)	0.00 (0.71) ***	0.00 (0.49)	0.00 (0.43)	0.00 (0.37)	0.00 (0.68) ***
Eggs	0.80 (1.54)	0.91 (1.06) *	0.60 (1.40)	1.20 (1.12) ***	1.32 (1.39)	1.44 (1.34) **
Dairy	0.61 (3.24)	5.98 (9.61) ***	2.03 (5.62)	4.98 (8.59) ***	0.00 (3.26)	4.43 (11.47) ***
Fats	0.75 (0.68)	1.03 (0.70) ***	1.21 (1.16)	1.57 (1.00) ***	1.51 (1.49)	2.30 (2.21) ***

^1^ IQR = Intraquartile Range; *** = significant at 0.001 level; ** = significant at 0.01 level; * = significant at 0.05 level.

**Table 3 ijerph-17-04739-t003:** Single and Multiple Food Group Models Relating Child HAZ to Previous Expenditures on Key Food Groups in (**a**) Ethiopia; (**b**) India; (**c**) Peru; and (**d**) Vietnam.

(a) Ethiopia—Expenditures at 5 Years with HAZ at 8 Years as the Dependent Variable—Predicting Future HAZ at Round 3									
Food Groups	Single Food Groups			Protein		Micronutrients		ASFs	
	Coefficient	*p*-Value	R-Sq	R-Sq: 0.0159		R-Sq: 0.0191		R-Sq: 0.0055	
				Coefficient	*p*-Value	Coefficient	*p*-Value	Coefficient	*p*-Value
Fruits and Vegetables	0.12	<0.001	0.0077			0.08	0.011		
Legumes	0.08	<0.001	0.0136	0.08	<0.001	0.07	<0.001		
Meat	0.02	0.046	0.0018	0.004	0.73	0.001	0.922	0.01	0.404
Eggs	0.19	0.007	0.0038	0.09	0.218	0.07	0.388	0.13	0.06
Dairy	0.04	0.006	0.004	0.03	0.094	0.02	0.28	0.03	0.086
Starches	0.02	<0.001	0.0123						
Fats	0.15	<0.001	0.0249						
Proteins	0.03	<0.001	0.0108						
Micronutrients	0.03	<0.001	0.0128						
Animal Source Foods	0.02	0.002	0.0049						
**Ethiopia—Expenditures at 8 Years with HAZ at 12 Years as the Dependent Variable**									
**Food Groups**	**Single Food Groups**			**Protein**		**Micronutrients**		**ASF**	
	**Coefficient**	***p*-Value**	**R-Sq**	**R-Sq: 0.0036**		**R-Sq: 0.0107**		**R-Sq: 0.0042**	
				**Coefficient**	***p*-Value**	**Coefficient**	***p*-Value**	**Coefficient**	***p*-Value**
Fruits and Vegetables	0.13	<0.001	0.0089			0.12	<0.001		
Legumes	0.01	0.623	−0.0004	0.005	0.826	−0.01	0.816		
Meat	0.04	0.002	0.0053	0.04	0.004	0.04	0.008	0.04	0.004
Eggs	0.07	0.278	0.0001	−0.005	0.942	−0.05	0.481	−0.004	0.796
Dairy	0.02	0.252	0.0002	0.004	0.816	−0.01	0.653	0.004	0.956
Starches	0.01	<0.001	0.0066						
Fats	0.16	<0.001	0.0183						
Proteins	0.02	0.009	0.0035						
Micronutrients	0.02	0.001	0.0055						
Animal Source Foods	0.02	0.006	0.0039						
**(b) India—Expenditures at 5 Years with HAZ at 8 Years as the Dependent Variable**									
**Food Groups**	**Single Food Groups**			**Protein**		**Micronutrient**		**ASF**	
	**Coefficient**	***p*-Value**	**R-Sq**	**R-Sq: 0.0419**		**R-Sq: 0.0416**		**R-Sq: 0.0429**	
				**Coefficient**	***p*-Value**	**Coefficient**	***p*-Value**	**Coefficient**	***p*-Value**
Fruits and Vegetables	0.02	0.051	0.0016			−0.01	0.5		
Legumes	−0.02	0.519	−0.0003	−0.04	0.147	−0.03	0.197		
Meat	0.03	0.003	0.0044	0.02	0.046	0.02	0.039	0.02	0.066
Eggs	0.09	0.166	0.0005	−0.03	0.692	−0.02	0.793	−0.04	0.599
Dairy	0.1	<0.001	0.0405	0.1	<0.001	0.1	<0.001	0.1	<0.001
Starches	0.0001	0.982	−0.0006						
Fats	0.11	<0.001	0.011						
Proteins	0.04	<0.001	0.0232						
Micronutrients	0.03	<0.001	0.0182						
Animal Source Foods	0.05	<0.001	0.0281						
**India—Expenditures at 8 Years with HAZ at 12 Years as the Dependent Variable**									
**Food Groups**	**Single Food Groups**			**Protein**		**Micronutrients**		**ASF**	
	**Coefficient**	***p*-Value**	**R-Sq**	**R-Sq: 0.048**		**R-Sq: 0.0479**		**R-Sq: 0.0467**	
				**Coefficient**	***p*-Value**	**Coefficient**	***p*-Value**	**Coefficient**	***p*-Value**
Fruits and Vegetables	0.04	<0.001	0.0069			0.01	0.365		
Legumes	−0.004	0.85	−0.0005	−0.04	0.068	−0.04	0.047		
Meat	0.02	0.119	0.0008	0.01	0.275	0.01	0.347	0.01	0.394
Eggs	0.14	0.035	0.002	0.01	0.909	0.001	0.986	0.001	0.988
Dairy	0.1	<0.001	0.0473	0.1	<0.001	0.1	<0.001	0.1	<0.001
Starches	0.02	<0.001	0.0061						
Fats	0.14	<0.001	0.0212						
Proteins	0.04	<0.001	0.0219						
Micronutrients	0.03	<0.001	0.0213						
Animal Source Foods	0.05	<0.001	0.0276						
**(c) Peru—Expenditures at 5 Years with HAZ at 8 Years as the Dependent Variable**									
**Food Groups**	**Single Food Groups**			**Protein**		**Micronutrient**		**ASF**	
	**Coefficient**	***p*-Value**	**R-Sq**	**R-Sq: 0.0758**		**R-Sq: 0.0774**		**R-Sq: 0.0737**	
				**Coefficient**	***p*-Value**	**Coefficient**	***p*-Value**	**Coefficient**	***p*-Value**
Fruits and Vegetables	0.06	<0.001	0.0334			0.02	0.047		
Legumes	0.01	0.59	−0.0004	−0.04	0.026	−0.04	0.016		
Meat	0.02	<0.001	0.0304	0.01	0.009	0.01	0.066	0.01	0.02
Eggs	0.09	<0.001	0.0095	−0.004	0.88	−0.01	0.675	−0.01	0.665
Dairy	0.06	<0.001	0.0718	0.05	<0.001	0.05	<0.001	0.05	<0.001
Starches	−0.001	0.785	−0.0005						
Fats	0.03	0.238	0.0002						
Proteins	0.02	<0.001	0.0544						
Micronutrients	0.02	<0.001	0.0569						
Animal Source Foods	0.02	<0.001	0.0582						
**Peru—Expenditures at 8 Years with HAZ at 12 Years as the Dependent Variable**									
**Food Groups**	**Single Food Groups**			**Protein**		**Micronutrient**		**ASF**	
	**Coefficient**	***p*-Value**	**R-Sq**	**R-Sq: 0.0747**		**R-Sq: 0.0775**		**R-Sq: 0.0634**	
				**Coefficient**	***p*-Value**	**Coefficient**	***p*-Value**	**Coefficient**	***p*-Value**
Fruits and Vegetables	0.06	<0.001	0.0292			0.02	0.013		
Legumes	−0.04	0.04	0.0019	−0.09	<0.001	−0.09	<0.001		
Meat	0.03	<0.001	0.0393	0.02	<0.001	0.02	<0.001	0.02	<0.001
Eggs	0.09	<0.001	0.0073	0.01	0.733	0.0001	0.995	−0.01	0.724
Dairy	0.06	<0.001	0.0529	0.05	<0.001	0.04	<0.001	0.05	<0.001
Starches	−0.01	0.027	0.0023						
Fats	0.02	0.359	−0.0001						
Proteins	0.02	<0.001	0.0506						
Micronutrients	0.02	<0.001	0.0538						
Animal Source Foods	0.02	<0.001	0.058						
**(d) Vietnam—Expenditures at 5 Years with HAZ at 8 Years as the Dependent Variable**									
**Food Groups**	**Single Food Groups**			**Protein**		**Micronutrient**		**ASF**	
	**Coefficient**	***p*-Value**	**R-Sq**	**R-Sq: 0.0932**		**R-Sq: 0.0972**		**R-Sq: 0.0937**	
				**Coefficient**	***p*-Value**	**Coefficient**	***p*-Value**	**Coefficient**	***p*-Value**
Fruits and Vegetables	0.07	<0.001	0.0584			0.02	0.004		
Legumes	0.13	<0.001	0.0065	0.01	0.734	0.01	0.793		
Meat	0.03	<0.001	0.0619	0.02	<0.001	0.02	<0.001	0.02	<0.001
Eggs	0.09	<0.001	0.0093	0.02	0.464	0.004	0.858	0.02	0.452
Dairy	0.05	<0.001	0.0765	0.04	<0.001	0.03	<0.001	0.04	<0.001
Starches	−0.01	0.364	−0.0001						
Fats	0.19	<0.001	0.0172						
Proteins	0.03	<0.001	0.0925						
Micronutrients	0.02	<0.001	0.097						
Animal Source Foods	0.03	<0.001	0.0923						
**Vietnam—Expenditures at 8 years with HAZ at 12 years as the Dependent Variable**									
**Food Groups**	**Single Food Groups**			**Protein**		**Micronutrient**		**ASF**	
	**Coefficient**	***p*-Value**	**R-Sq**	**R-Sq: 0.0985**		**R-Sq: 0.1008**		**R-Sq: 0.0986**	
				**Coefficient**	***p*-Value**	**Coefficient**	***p*-Value**	**Coefficient**	***p*-Value**
Fruits and Vegetables	0.06	<0.001	0.0439			0.02	0.025		
Legumes	0.13	0.008	0.0037	0.04	0.369	0.03	0.484		
Meat	0.04	<0.001	0.0817	0.03	<0.001	0.03	<0.001	0.03	<0.001
Eggs	0.21	<0.001	0.0458	0.13	<0.001	0.12	<0.001	0.13	<0.001
Dairy	0.03	<0.001	0.0252	0.01	0.107	0.005	0.325	0.01	0.082
Starches	−0.004	0.505	−0.0003						
Fats	0.06	0.015	0.003						
Proteins	0.03	<0.001	0.0822						
Micronutrients	0.02	<0.001	0.0856						
Animal Source Foods	0.03	<0.001	0.0818						

**Table 4 ijerph-17-04739-t004:** Models of Child HAZ Assessing influence of Past Household Fats Expenditures in (**a**) Ethiopia, (**b**) India, (**c**) Peru and Past Household Starches Expenditures in (**d**) Vietnam.

(a) Ethiopia	Expenditures at 5 Years Associated with HAZ at 8 Years				Expenditures at 8 Years Associated with HAZ at 12 Years			
Fats	Model 1a		Model 2a		Model 1b		Model 2b	
Model Adjusted R-Squared	0.0249		0.0629		0.0183		0.0575	
	β	*p*-Value	β	*p*-Value	β	*p*-Value	β	*p*-Value
Fats	0.15	<0.001	0.03	0.291	0.16	<0.001	0.02	0.627
*Individual Variable*								
Female			0.13	0.006			−0.02	0.673
*Community Variable*								
Rural/Urban Status			−0.19	0.001			−0.30	<0.001
*Household Variable*								
Total Food Expenditures			0.01	0.01			0.003	0.204
*Maternal Education*								
No Schooling			Reference				Reference	
1–6 years			0.14	0.017			0.15	0.008
7–12 years			0.2	0.024			0.18	0.036
12+ years			−0.02	0.903			0.12	0.522
*Paternal Education*								
No Schooling			Reference				Reference	
1–6 years			0.14	0.016			0.09	0.125
7–12 years			0.19	0.018				
12+ years			0.26	0.039				
**(b) India**	**Expenditures at 5 Years Associated with HAZ at 8 Years**				**Expenditures at 8 Years Associated with HAZ at 12 Years**			
**Fats**	**Model 1a**		**Model 2a**		**Model 1b**		**Model 2b**	
**Model Adjusted R-Squared**	**0.011**		**0.1215**		**0.0212**		**0.1008**	
	**β**	***p*-Value**	**β**	***p*-Value**	**β**	***p*-Value**	**β**	***p*-Value**
Fats	0.11	<0.001	0.07	0.015	0.14	<0.001	0.06	0.025
*Individual Variable*								
Female			0.12	0.005			0.04	0.376
*Community Variable*								
Rural/Urban Status			−0.40	<0.001			−0.34	<0.001
*Household Variable*								
Total Food Expenditures			0.001	0.795			0.001	0.572
*Maternal Education*								
No Schooling			Reference				Reference	
1–6 years			0.17	0.005			0.11	0.075
7–12 years			0.22	<0.001			0.12	0.063
12+ years			0.6	<0.001			0.48	0.003
*Paternal Education*								
No Schooling			Reference				Reference	
1–6 years			0.11	0.071			0.17	0.005
7–12 years			0.18	0.003			0.29	<0.001
12+ years			0.37	<0.001			0.4	<0.001
**(c) Peru**	**Expenditures at 5 Years Associated with HAZ at 8 Years**				**Expenditures at 8 Years Associated with HAZ at 12 Years**			
**Fats**	**Model 1a**		**Model 2a**		**Model 1b**		**Model 2b**	
**Model Adjusted R-Squared**	**0.0002**		**0.2194**		**−0.0001**		**0.2003**	
	**β**	***p*-Value**	**β**	***p*-Value**	**β**	***p*-Value**	**β**	***p*-Value**
Fats	0.03	0.238	0.05	0.042	0.02	0.359	0.04	0.101
*Individual Variable*								
Female			0.03	0.511			−0.06	0.206
*Community Variable*								
Rural/Urban Status			−0.47	<0.001			−0.47	<0.001
*Household Variable*								
Total Food Expenditures			−0.0003	0.727			0.003	<0.001
*Maternal Education*								
No Schooling			Reference				Reference	
1–6 years			0.26	0.003			0.24	0.013
7–12 years			0.64	<0.001			0.6	<0.001
12+ years			0.84	<0.001			0.72	<0.001
*Paternal Education*								
No Schooling			Reference				Reference	
1–6 years			0.11	0.57			-0.001	0.997
7–12 years			0.2	0.296			0.16	0.443
12+ years			0.35	0.085			0.37	0.094
**(d) Vietnam**	**Expenditures at 5 Years Associated with HAZ at 8 Years**				**Expenditures at 8 Years Associated with HAZ at 12 Years**			
**Starches**	**Model 1a**		**Model 2a**		**Model 1b**		**Model 2b**	
**Model Adjusted R-Squared**	**−0.0001**		**0.2013**		**−0.0003**		**0.175**	
	**β**	***p*-Value**	**β**	***p*-Value**	**β**	***p*-Value**	**Β**	***p*-Value**
Starches	−0.01	0.364	−0.01	0.014	−0.004	0.505	−0.01	0.047
*Individual Variable*								
Female			0.09	0.056			0.01	0.852
*Community Variable*								
Rural/Urban Status			−0.36	<0.001			−0.32	<0.001
*Household Variable*								
Total Food Expenditures			0.01	<0.001			0.01	<0.001
*Maternal Education*								
No Schooling			Reference				Reference	
1–6 years			0.55	<0.001			0.65	<0.001
7–12 years			0.64	<0.001			0.68	<0.001
12+ years			0.96	<0.001			0.92	<0.001
*Paternal Education*								
No Schooling			Reference				Reference	
1–6 years			0.32	0.002			0.22	0.061
7–12 years			0.39	<0.001			0.37	0.003
12+ years			0.5	0.001			0.54	0.001

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
