# Peer review of "Are Household Expenditures on Food Groups Associated with Children’s Future Heights in Ethiopia, India, Peru, and Vietnam?"

_ijerph, 2020, doi:10.3390/ijerph17134739_

Round 1

Reviewer 1 Report

Thanks. Please see comments as attached

Author Response

Reviewer 1:

In this study Weingarten et al used data on 6,993 children born around 2001 from Ethiopia, India, Peru, and Vietnam collected at ages 5, 8, and 12 years from the Young Lives younger cohort to assess the longitudinal associations between expenditures on key food groups at 5y and 8y (including starches, fats, legumes, meat, fruits and vegetables, eggs, dairy, animal source food, proteins, and micronutrients) and child HAZ at ages 8y and 12y, respectively.

Although tremendous heterogeneity and limitations are exist with this study, however, still study like this can serve potentially beneficial knowledge for the community regarding children health with common food groups at different location. Overall, study is well written and detailed. Following are some information that author should incorporate in the updated manuscript.

  1. Author should add the pre-existed health condition of the children, for example, were there child already suffered with stunting, obesity, or wasting?

We agree that pre-existing health of the child, including conditions such as stunting, impact child growth.  At the same time, the research question addressed in the manuscript is the relationship between food group expenditures and future HAZ as the reviewer summarizes in the first paragraph above, rather than the best predictors of future HAZ overall. Determining the role of stunting—a biological covariate--and subsequent HAZ is a different research question which would be valuable to examine in a separate set of analyses that seek to identify the many potential predictors of stunting, but not in this study.

  1. What is the impact on healthy versus malnourished with fats or starches consume and expenditure? These question answer can lead us to know how observation like fats or starches involved!

We agree that these are interesting questions that can be addressed with our data.  However, we see them as a separate, and future analysis.

Reviewer 2 Report

This is an interesting study exploring the associations between household food expenditures and child HAZ in several LMICs using longitudinal data. I think the quality of this manuscript is overall good. However, I have several concerns for the authors to be addressed before the consideration of publication.

  1. I think the current title is more of a statement of a result than a title. Considering replacing it. 
  2. Parental heights are very important in predicting child's HAZ. Although the research subjects are the impacts of household food expenditures on child's HAZ, the authors should include parental heights as key control variables.
  3. The authors mainly estimated the lagged impact of household food expenditures 3 or 4 years ago on child height, have they considered the influence of the current household food expenditures? In my opinion, current household food expenditures should at least be listed as control variables to check the robustness of these associations.
  4. I suggest the authors to update their references. Clearly many of them are vey old.

Reviewer 3 Report

Dear Author, 

It is a well written and interesting manuscript. Some comments/suggesting on how to improve: 

Line 40 - please use more recent data 

Introduction - I suggest to rewrite the introduction and more focus on the aim ".... how different food expenditures relate to 70 children’s subsequent nutritional status". For example are there any previous data on a similar association?; the way the introduction is written suggest that you are going to validate a new method to examine the effects of child diet on nutritional status. 

Lines 70-78 - can go to the methods section 

Author Response

Response to Reviewer 3: 

Dear Author,

It is a well written and interesting manuscript. Some comments/suggesting on how to improve:

Line 40 - please use more recent data

We appreciate the suggestion, and have added a current estimate.

Introduction - I suggest to rewrite the introduction and more focus on the aim ".... how different food expenditures relate to children’s subsequent nutritional status". For example are there any previous data on a similar association?; the way the introduction is written suggest that you are going to validate a new method to examine the effects of child diet on nutritional status.

Thanks for the suggestion.  We have revised the introduction to try to give more focus to our basic research question.

Lines 70-78 - can go to the methods section

We have followed this recommendation.

Reviewer 4 Report

Title: In Ethiopia, India, and Peru, household expenditures on fats are more strongly associated with children’s  future heights than expenditures on other food groups

ROUND 1

This paper presents and discusses a longitudinal study using data coming obtained in the YL study and data from HECS for 4 different countries (Ethiopia, India, Peru, and Vietnam).

The paper is quite interesting, especially for researchers interested on the factors that condition stunting in children from developing countries.

After a carefully reading of the paper, these are my commentaries and suggestions:

  1. LINE 30: height-for-age-Z score (HAZ).
  2. LINE 32-35: be more concise on what was the main conclusion.
  3. LINE 63: I don’t agree with this so general statement. You cited 3 references supporting this but one (11) is a preliminary study and the other two (13 and 14) are studies made in developing countries. In developed countries, differences between HCES and R24h can be high because food waste is much more frequent. Additionally, results obtained with both methods can agree in some food groups but can be quite different in others (https://pubmed.ncbi.nlm.nih.gov/15975190/). So, please, explain more precisely in which cases HBS and 24HR DR could render a good association.
  4. LINE 64-65: What do you exactly mean with this statement? It is confusing. HCES and 24 HR DR don’t include an assessment of child nutritional status. Because you are referring in a generalist way to HBS and 24 HR DR this statement is not correct. Be more precise. Additionally, families/children included in the YL also constitute the sample for the HECS?
  5. Figure 1: Define the acronym HAZ and YL in the figure legend; figures and tables should be self-explanatory. Additionally, this figure and the associated explanations (conceptual framework) should be included in the section Method design, not in the Introduction. In general, define acronyms used in figures (what does mean IQR in Figure 2?).
  6. LINES 86-88: You should state here clearly if the subsequent data collection is made on the same individuals.
  7. LINES 105-107: So, I understand you don’t include the production of food at home (orchard, fruit grove, hen house, etc.), right?
  8. FILE 155-160: Include this information at the end of the section “Study Indicators” without referring to Figure 2 and explaining that this data was used to check the heterogeneity in the country-level food energy and macronutrient availability. My suggestion comes from the fact that study indicators should be assessed taking in consideration this heterogeneity.
  9. LINES 188-189: percentages don’t remain “relatively” constant; there is a clear difference between Ethiopia and Vietnam; you can’t say that.
  10. Have you investigated possible collinearities between some of the independent variables (e.g. total food expenditure and schooling)?
  11. LINES 442-443: …because its high energy content; this is obvious, but you should mention it.
  12. LINES 443-445: … and because some fats CONTAINS liposoluble vitamins! Fat from milk and milk products contains vitamin A and D, some vegetables fats contains considerable amounts of betacarotene, and fatty fish is rich in both vitamins. It is a pity that you couldn’t aggregate fish, milk and vegetable fats expenditures in a group.

Round 2

Reviewer 2 Report

I think most of the comments are reasonably addressed.